# Fear perception as a function of hemisphere- and time-specific dynamics in the medial temporal lobes

Enya M. Weidner [1] ✉, Lea Marie Reisch[1,2], Malena Mielke[1], Christian G. Bien[2] & Johanna Kissler [1]

The medial temporal lobes (mTL) are thought to enhance visual processing of fearful faces, yet the underlying mechanisms remain underspecified. To fill this gap, we recorded and compared event-related potentials (ERPs) and stimulus-induced gamma-band activity (GBA) from 36 patients with left- or right-hemispheric antero-medial temporal lobe resections including the amygdala (lTLR/rTLR) and 18 healthy controls. Only rTLR patients were found to lack fear-neutral differentiation in early P1 amplitudes (~100 ms) and exhibited heightened GBA for neutral faces over ipsi-resectional occipito-temporal areas (95–300 ms). lTLR patients showed strongest emotion differentiation in ERP components beyond the P1. Therefore, the right mTL, potentially particularly the amygdala, appears to support rapid attentional shifts toward fear and to coordinate fear-neutral differentiation in GBA. Conversely, the left mTL seems to down-regulate fear responses. These results reveal complementary, lateralized, and time-specific roles of the medial temporal lobes in fear processing, thereby refining models of emotional vision.

The human visual system prioritizes facial expressions as they are essential for communication and interaction[1–3], above all when they display fear[4]. Medial temporal lobe (mTL) structures, particularly the amygdala, are thought to rapidly facilitate the allocation of visual attention to emotionally salient faces through bi-directional connections to the visual system[5,6]. Although a famous patient with bilateral amygdala destruction has been shown to be severely impaired when processing fearful faces[7], some functional MRI (fMRI) studies reported preserved visual discrimination of fearful from neutral faces in individuals with unilateral[8] and potentially even bilateral amygdala lesions[9]. This suggests the existence of multiple pathways of visual emotion processing that are not yet well-characterized[10,11]. Crucially, it is largely unknown when and how the mTL drives processing enhancements for fearful faces, whether the hemispheres differ in their contribution, and when emotion differentiation might occur independently of the mTL. Current models on emotional vision[5,11] lack characterization of the role of the mTL across time. Such insights could also be relevant to advance clinical applications of affective neuroscience[12–15].

Electrophysiological studies on patients with mTL damage or resections help delineate its role during fear perception with high temporal precision. For instance, studies on components of event-related potentials (ERPs) are well-suited to characterize the signal strength of cortical afferents and efferents over time[16,17]. Given that the functional significance of many ERP components is known, it is also possible to assign any disruptions to

changes in perceptual and cognitive processes. However, so far, sparse small-n studies have yielded inconclusive findings: An older study associated primarily left amygdala sclerosis (six out of seven studied patients) with lacking differential responses to fearful faces in early visual attention around 100 ms (P1 component) and later, around 600 ms (P600 component), in processes associated with memory encoding and stimulus evaluation[18]. By contrast, a recent report[19] observed absence of the often reported N1 component enhancement for fearful faces[4,20] in six patients with right but not eight with left temporal lobe resections (rTLR/lTLR, respectively), suggesting an impact of rTLR on the structural encoding of fearful faces and no effect of lTLR[4,21]. Thus, evidence is contradictory as to when exactly the mTL, potentially particularly the amygdala, facilitates fear perception as operationalized by scalp ERPs. Furthermore, the lateralization of mTL and specifically amygdala functions has long been debated[22–25], partly due to the lack of systematic comparisons between effects of right and left mTL lesions.

Oscillatory gamma-band activity (GBA, 35–90 Hz) allows insights into intra-network computations during information processing and integration. It is thought to encode how efficiently internal neural representations are adapted to certain stimulus attributes[26–29]. A well-adapted system results in stronger within-system GBA synchronization, indicating successful reduction of activity redundancies, so-called sparsification through predictive coding[26]. Fittingly, Li and Keil[10] summarize evidence that assumes

[1]Department of Psychology, Bielefeld University, Bielefeld, Germany. [2]Department of Epileptology, Krankenhaus Mara, Bethel Epilepsy Center, Medical School OWL, Bielefeld University, Bielefeld, Germany. ✉e-mail: enya.weidner@uni-bielefeld.de

visual cortical sparsification as one main contributor to fear processing. Indeed, visual cortical GBA has been associated with fear discrimination[30–32]. However, although a role for GBA is sometimes implicitly assumed[32], to our knowledge, there are no studies investigating the effect of mTL lesions on GBA correlates of emotional face processing.

Therefore, the present study explores the time- and laterality-specific contributions of the mTL to the emotion-sensitive ERP components, P1, N1, early posterior negativity (EPN), and late positive potential (LPP)[4,18,19,33–35], as well as to oscillatory GBA[32,36–39]. ERPs and GBA are complimentary and temporally highly resolved neural signatures of emotional face processing. We derived these markers from a comparatively large and well characterized sample of patients with unilateral temporal lobe resections (18 lTLR and 18 rTLR, Fig. 1) and a case-matched healthy control group (HC). Participants were presented with a series of fearful and neutral facial expressions. Building on previous research[19], we expected that rTLR patients would exhibit reduced emotion differentiation in early ERP components. Given the sparse data on emotion differentiation in lTLR patients, we did not formulate directional a priori hypotheses for this group. Nevertheless, based on previous findings of altered fear processing in individuals with left-hemispheric amygdala lesions[18], we anticipated that their responses would deviate from the overall group mean. Our primary focus is on electrophysiology data, but we also assessed subjective stimulus appraisal and recognition memory for faces in the three groups of participants.

## Results

### Higher arousal and negative valence for fearful faces across groups

Self-reports showed higher arousal for fearful compared to neutral faces ($M_{fear} = 4.145$, $SD_{fear} = 1.370$; $M_{neutral} = 2.991$, $SD_{neutral} = 0.933$; $\beta = -1.035$, $p = 2.67e^{-06}$). Valence ratings were significantly lower for fearful than for neutral faces ($M_{fear} = 2.798$, $SD_{fear} = 0.740$; $M_{neutral} = 3.717$, $SD_{neutral} = 0.700$; $\beta = 0.900$, $p = 3.33e^{-07}$). Groups did not differ regarding their arousal or valence ratings (Supplementary Fig. 1a, Supplementary Tables 1 and 2).

### Reduced face recognition in rTLR and general recognition bias for fearful faces

Overall recognition scores (discrimination index, DI) were worse in rTLR (DI: $M_{rTLR} = 0.132$, $SD_{rTLR} = 0.150$; $M_{lTLR} = 0.230$, $SD_{lTLR} = 0.189$; $M_{HC} = 0.227$, $SD_{HC} = 0.206$; $\beta_{Group} = -0.065$, $p = 0.012$). Across groups, recognition bias was higher for fearful than neutral faces (Bias: $M_{fearful} = 0.494$, $SD_{fearful} = 0.181$, $M_{neutral} = 0.365$, $SD_{neutral} = 0.178$; $\beta_{Emotion} = 0.644$, $p = 0.0005$, Supplementary Fig. 1b, Supplementary

Tables 3 and 4). There was no *Group × Emotion* interaction on recognition accuracy.

### ERPs reveal attenuated early fear signals in rTLR and a sustained increase of fear differentiation in lTLR

Our main interest was to characterize the temporal dynamics of fear processing between and within groups. Therefore, linear mixed models (LMMs) were applied to single-trial amplitude values of four ERP components that are typically sensitive to visual and higher cognitive emotion processing. Additionally, we controlled for clinical and resection-specific factors in separate control models. Finally, we assessed associations of electrophysiological and behavioral measures. For brevity, we only report contrasts that relate to our hypotheses in the main text. These are also illustrated in Fig. 2. Supplementary Tables 5 and 6 document all model coefficients.

Importantly, while we found an overall effect of *Emotion* ($\beta_{Emotion} = -0.156$, $p < 2e^{-16}$), this interacted with the *Group* and *ERP component*. Patient groups exhibited distinct emotion differentiation depending on the ERP component: In the P1, rTLR patients exhibited the least emotion differentiation ($\beta_{Emotion \times Group (rTLR) \times ERP} = -0.081$, $p = 0.025$). For HC but not rTLR patients, P1 amplitudes were higher in response to fearful than neutral faces (HC: $\beta_{fearful - neutral} = 0.260$, $p = 0.018$, $d = 0.494$; rTLR: $\beta_{fearful - neutral} = -0.097$, $p = 0.356$, $d = -0.234$, Supplementary Table 5). For lTLR patients, higher P1 amplitudes in response to fearful than neutral faces were present in a forward-shifted electrode cluster (lTLR: $\beta_{fearful - neutral} = 0.106$, $p = 0.050$, $d = 0.410$, Supplementary Fig. 2) but not in the a priori defined cluster ($\beta_{fearful - neutral} = 0.119$, $p = 0.277$, $d = 0.185$). The shifted cluster was chosen based on post-hoc assessment of the differential (fearful-neutral) P1 topography in lTLR patients. No other group showed significant emotion differentiation in the shifted cluster (HC: $\beta_{fearful - neutral} = 0.062$, $p = 0.270$, $d = 0.334$, rTLR: $\beta_{fearful - neutral} = -0.023$, $p = 0.696$, $d = -0.104$ (Supplementary Table 6).

Beyond the P1, all groups showed an emotion differentiation in favor of fear, i.e., lower amplitudes in response to fearful than neutral faces in the N1 and EPN component and higher amplitudes in response to fearful than neutral faces in the LPP component (all $ps < 0.009$, all $ds > 0.604$; Supplementary Table 6). However, lTLR patients exhibited larger emotion differentiation than the unweighted average across all groups in the N1 as well as LPP component (N1: $\beta_{Emotion \times Group (lTLR) \times ERP} = -0.081$, $p = 0.040$; LPP: $\beta_{Emotion \times Group (lTLR) \times ERP} = 0.094$, $p = 0.017$). A separate LMM for the EPN (Supplementary Table 7) confirmed that this was, as a trend, also present in the EPN ($\beta_{Emotion \times Group (lTLR)} = -0.080$, $p = 0.084$). Control models revealed that larger LPP amplitudes in response to fearful than neutral faces in rTLR patients was associated with smaller fusiform resection volume ($\beta_{Group (rTLR) \times ERP} = -0.032$, $p = 0.026$). This was not observed for the other

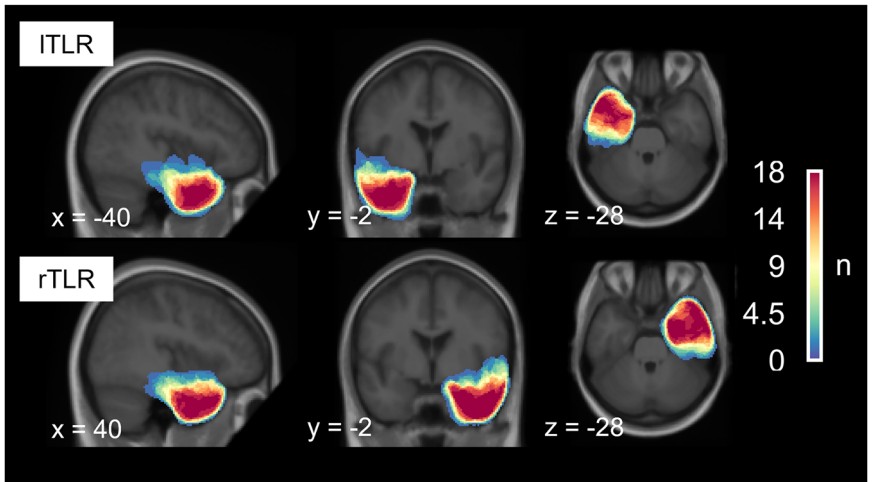

**Fig. 1 | Extent of the resections.** Overlap of temporal lobe resections, which consisted of anteromedial (including the hippocampus, $n = 13$) and apical (hippocampus sparing, $n = 5$) temporal lobe resections in the lTLR patients (top row) and exclusively of anteromedial temporal lobe resections ($n = 18$) in all rTLR patients ($n = 18$, bottom row). Images are centered at the mean overlap maxima of the resections. *Abbreviations*: lTLR left temporal lobe resection, rTLR right temporal lobe resection.

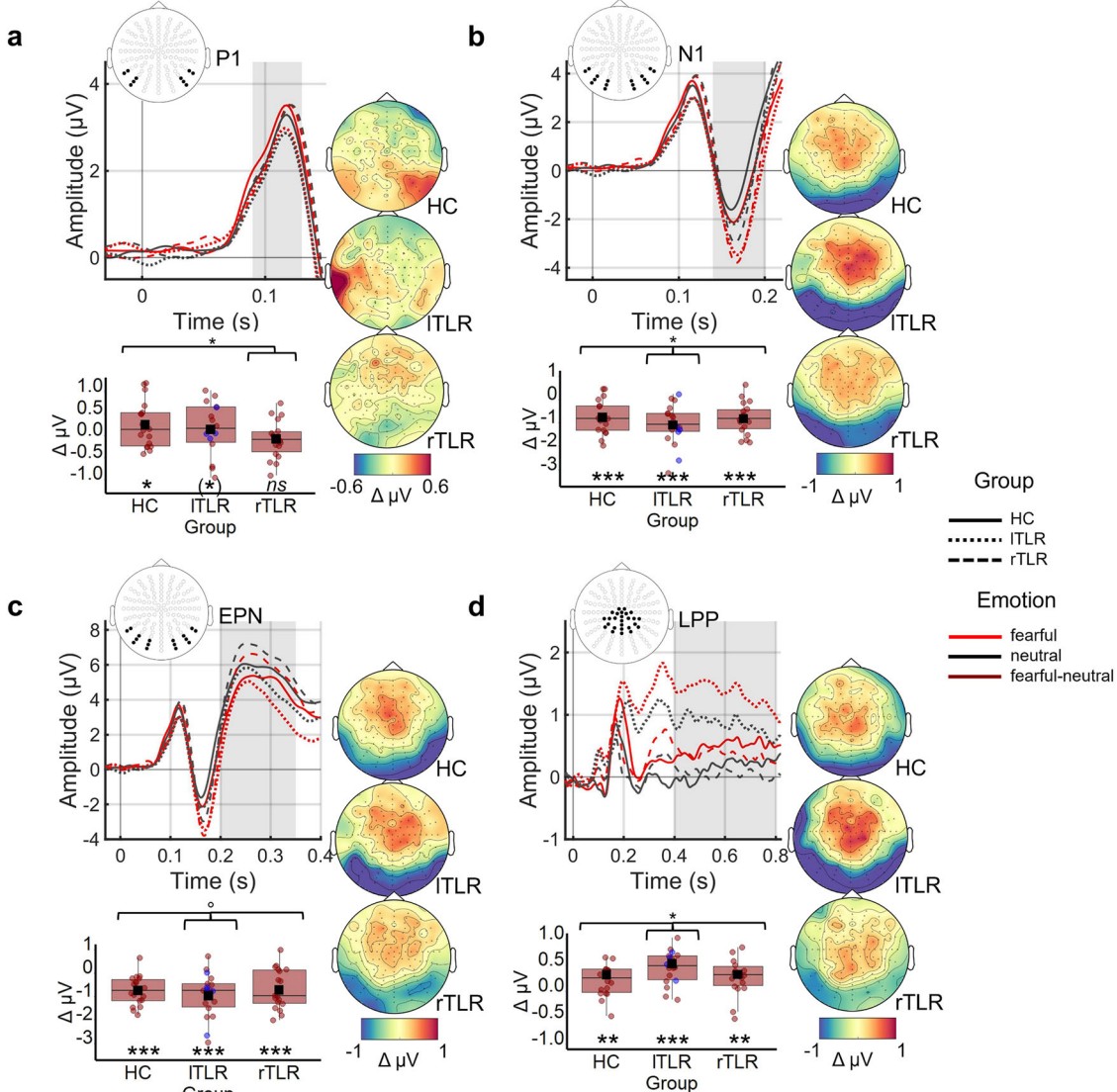

**Fig. 2 | ERP components and the difference between fearful and neutral faces for individual groups. a** Data and analysis for the P1. rTLR patients exhibited the least emotion differentiation ($\beta_{\text{Emotion} \times \text{Group (rTLR)} \times \text{ERP}} = -0.081$, $p = 0.025$). **b** Data and analysis for the N1. lTLR patients exhibited larger emotion differentiation than the unweighted average across all groups ($\beta_{\text{Emotion} \times \text{Group (lTLR)} \times \text{ERP}} = -0.081$, $p = 0.040$). **c** Data and analysis for the EPN. As a trend, lTLR patients exhibited larger emotion differentiation than the unweighted average across all groups ($\beta_{\text{Emotion} \times \text{Group (lTLR)}} = -0.080$, $p = 0.084$). **d** Data and analysis for the LPP. lTLR patients exhibited larger emotion differentiation than the unweighted average across all groups ($\beta_{\text{Emotion} \times \text{Group (lTLR)} \times \text{ERP}} = 0.094$, $p = 0.017$). ERP curves depict mean amplitudes of the marked channel clusters for each group ($n = 18$ per group). The grey area indicates the time-window in which the ERP component was extracted from the channels marked in black. Difference topographies (fearful-neutral) are averaged across time-points within the respective time-window. Boxplots show differential (fearful-neutral) amplitudes extracted from the channel clusters, averaged across time. Whiskers indicate the interquartile range. The bold horizontal line indicates the distribution median. Dots indicate single-subject values. Black squares indicate the distribution mean. lTLR patients with apical resections (spared hippocampus, $n = 5$) are marked in blue. Brackets mark significant comparisons that specify significant differences and interactive effects of *Group* and *Emotion* (*$p \leq 0.05$, **$p \leq 0.01$, ***$p \leq 0.001$). Since significant emotion differentiation in lTLR patients was shifted forwards ($\beta_{\text{fearful - neutral}} = 0.106$, $p = 0.050$, $d = 0.410$; Supplementary Fig. 2) and not found in the a priori defined cluster ($\beta_{\text{fearful - neutral}} = 0.119$, $p = 0.277$, $d = 0.185$), the significance marking is depicted in brackets. *Abbreviations*: HC healthy controls, lTLR left temporal lobe resection, ns not significant, rTLR right temporal lobe resection.

components. The lTLR group showed no effects of resection volume on electrophysiological emotion differentiation (Supplementary Table 10).

### Brain-behavior associations

For the HC and rTLR group, a larger difference of EPN amplitudes in favor of fearful faces was indicative of an enhanced memory bias for fearful relative to neutral faces (HC: $\beta_{\text{fearful-neutral}} = -0.177$, $t_{(35)} = -1.852$, $p = 0.072$, rTLR: $\beta_{\text{fearful-neutral}} = -0.161$, $t_{(35)} = -2.099$, $p = 0.043$). This was not present in lTLR patients ($\beta_{\text{fearful-neutral}} = 0.103$, $t_{(35)} = 1.321$, $p = 0.195$), as also reflected in the *Group × EPN* interaction ($\beta_{\text{Group (lTLR)} \times \text{EPN}} = 0.182$, $p = 0.011$; Supplementary Fig. 4,

Supplementary Table 9). Furthermore, only for rTLR patients was there an association between LPP emotion differentiation and the differential recognition bias: When the difference in LPP amplitudes between fearful and neutral faces decreased, the recognition bias increased for fearful relative to neutral faces ($\beta_{\text{Group (rTLR)} \times \text{LPP}} = -0.201$, $p = 0.030$; Supplementary Fig. 4, Supplementary Table 9; rTLR: $\beta_{\text{fearful-neutral}} = -0.267$, $t_{(35)} = -2.333$, $p = 0.026$, HC: $\beta_{\text{fearful-neutral}} = 0.061$, $t_{(35)} = 0.880$, $p = 0.385$, lTLR: $\beta_{\text{fearful-neutral}} = 0.008$, $t_{(35)} = 0.102$, $p = 0.920$). Additionally, the analyses revealed a significant association between N1 emotion differentiation and arousal ratings across all groups: The higher the N1 amplitude difference in favor of fearful faces, the higher the difference in

arousal ratings between fearful and neutral faces ($\beta_{N1} = -0.874$, $p = 0.008$). No significant effects were found for the valence ratings (all $p$s > 0.100).

## GBA reveals altered cortical organization in rTLR

With pointwise cluster-based permutation tests of fixed-effects $t$-values, we identified time- and frequency-windows of GBA (35–90 Hz)[40] alterations during emotion processing within and between the groups. A main effect of *Emotion* was found over left occipital contacts from 85 – 195 ms in a frequency range of 70-85 Hz, driven by higher GBA power in response to neutral than fearful faces (Supplementary Fig. 3). Critically, the fearful-neutral differentiation at right occipital contacts differed significantly for rTLR patients compared to the unweighted average across all groups from 95 to 300 ms in a frequency range of 60–80 Hz (summed cluster $t_{\text{Group (rTLR)} \times \text{Emotion}} = -1421.000$, $p = 0.0002$). This difference was driven by higher posterior GBA power in response to neutral than fearful faces in rTLR patients only ($\beta_{\text{fearful-neutral}} = -0.048$, $t$-ratio $= -3.875$, $p = 0.0001$, $d = -0.856$, Figs. 3, 4, Supplementary Table 8, Supplementary Fig. 3).

The control analyses (Supplementary Table 11) suggested no significant effect of the control factors on the *Group (rTLR) × Emotion* interaction. Likewise, we found no significant associations between GBA power and behavioral responses (Supplementary Table 9). A summary of the decisive interactions in ERP components and GBA is reported in Table 1.

## Discussion

Present electrophysiological data reveal distinct temporal profiles of emotional face processing in rTLR and lTLR patients. Specifically, rTLR patients exhibited reduced P1 amplitudes to fearful compared to neutral faces, while also demonstrating increased GBA in response to neutral versus fearful faces over ipsi-resectional, i.e., right occipito-temporal, areas. By contrast, lTLR

patients showed enhanced fear differentiation in ERP components beyond the P1 and no distinct pattern in GBA.

The P1 results imply that rTLR patients fail to effectively tune initial perception towards fearful faces. The P1 amplitude in response to emotion has been previously associated with amygdala blood flow[33] and is mostly generated by input to the visual cortex[41]. Thus, specifically the feedback from the right amygdala to visual cortices appears to be absent in rTLR patients[5]. Correspondingly, emotion differentiation in the P1 was previously reported to be preserved in patients with mTL lesions sparing the amygdala[18]. However, this previous study reported a lack of P1 enhancement by fearful faces in six patients with left amygdala sclerosis, whereas our larger sample indicates this to be due to absence of the right one. One reason for this discrepancy might be sclerosis-induced reorganization or functional disruptions in the previous report[42]. In the present sample, according to histological exams, amygdalae were structurally intact at resection. Furthermore, we found no evidence of the influence of hippocampal or fusiform resection extent on P1 emotion differentiation in rTLR patients. Thus, particularly efferents from the right amygdala to the visual cortex may be crucial to heightening the competitive strength of fearful faces in early, pre-conscious visual attention. The right amygdala likely evaluates rudimentary perceptual cues from subcortical, magnocellular pathways in a fast first-sweep of emotion detection and subsequently inputs information to the visual cortex[6,43]. This could also lead to absent orienting responses towards diagnostic facial features[44] and impaired emotion recognition in low-intensity and ambiguous facial expressions[8]. Interestingly, absent emotion differentiation in the P1 in rTLR patients is seemingly not limited to facial stimuli, as it was also found for more complex emotional scenes[35] and more abstract written words[45]. Thus, early emotion discrimination might rely on the right amygdala across stimulus modalities. This might specifically underlie rapid, low resolution thalamo-amygdalar fear cues[33,46]. Since descriptively, ERP difference potentials (fearful minus neutral), while

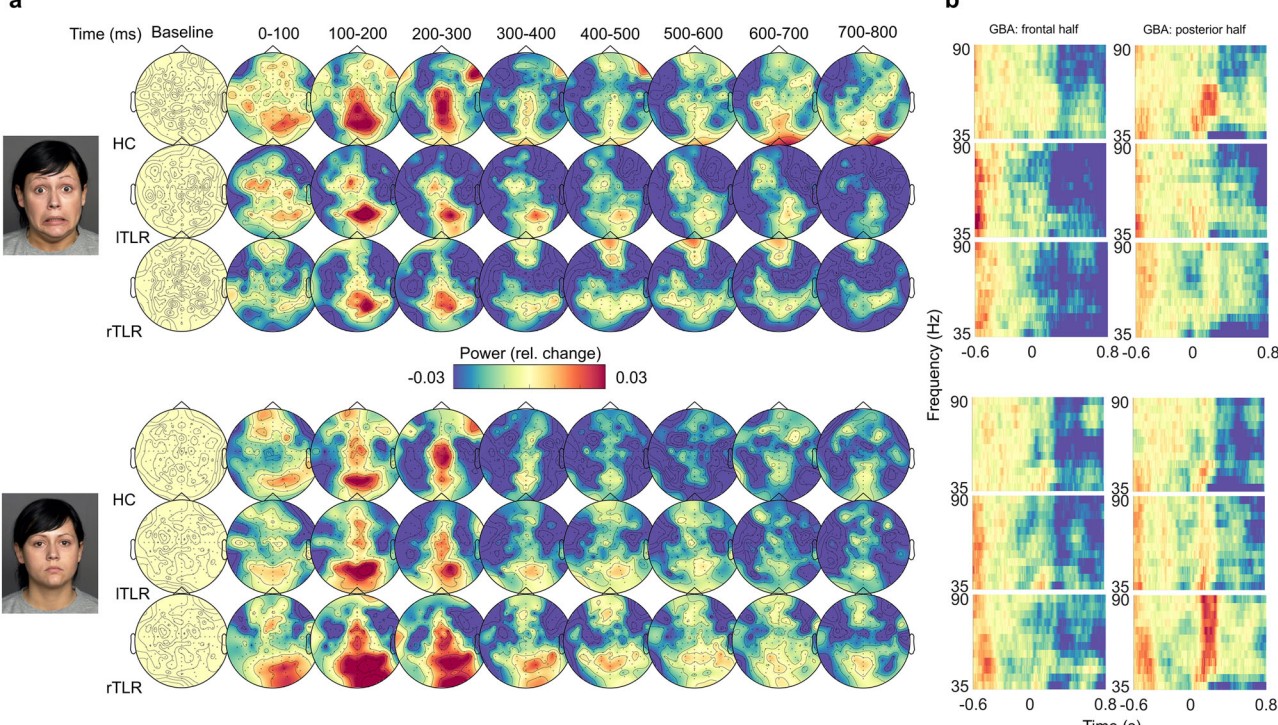

**Fig. 3 | Changes in GBA power in response to fearful and neutral faces in each group. a** Grand-average time-series of GBA power topographies (35–90 Hz). Data per group ($n = 18$ per group) are averaged across the indicated time-points and frequencies. **b** Grand-average time-frequency plots, averaged across the frontal and posterior half of the scalp electrodes. Face images are derived from the FACES database[82] and printed with previous permission. *Abbreviations*: HC healthy controls, lTLR left temporal lobe resection, rTLR right temporal lobe resection.

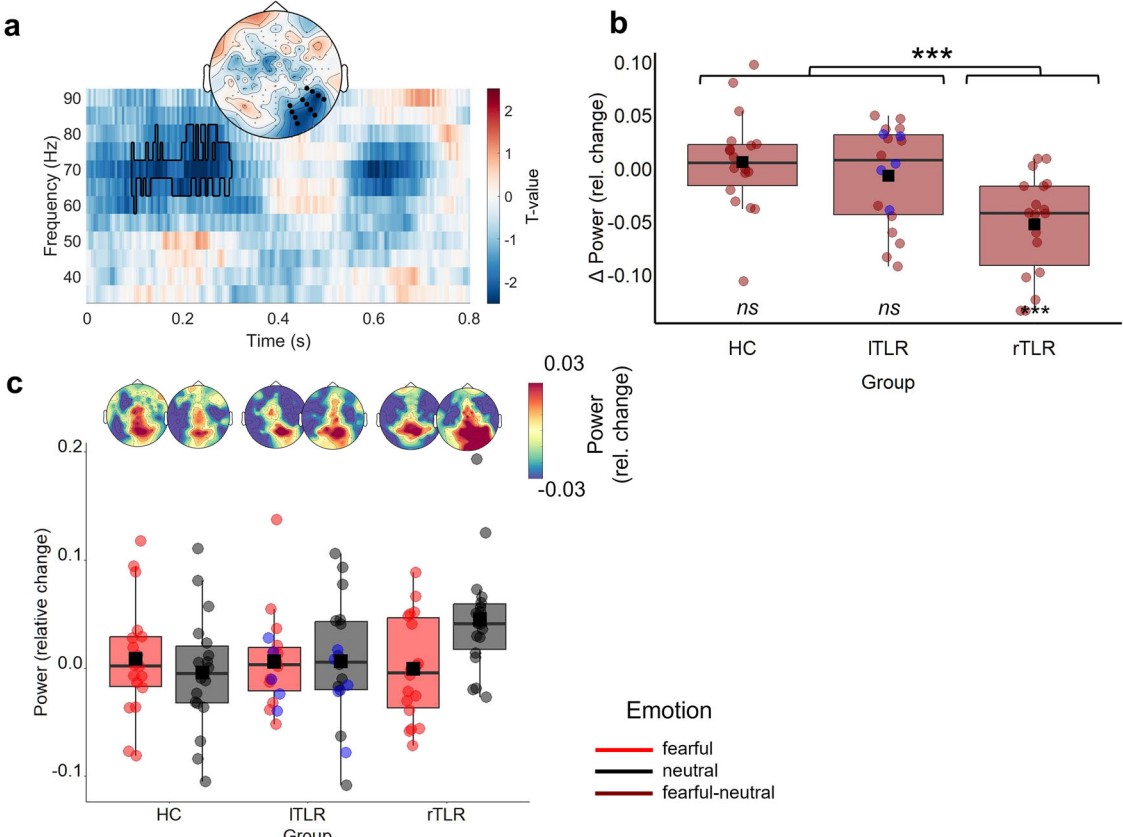

**Fig. 4 | Differential GBA fear responses between groups. a** Distribution of *t*-values where fearful-neutral differentiation varied between rTLR patients and the unweighted average across all groups (summed cluster $t_{\text{Group (rTLR)} \times \text{Emotion}} = -1421.000$, $p = 0.0002$; $n = 18$ per group). The topography is averaged across time-points (95–300 ms) and frequencies (60–80 Hz) of significant effects. Time-frequency plots are averaged across channels. Significant channels and time-windows are outlined or marked in black. **b** Distribution of differential GBA power within the cluster of the significant *Group (rTLR) × Emotion* interaction. **c** Distribution of power values per group and condition extracted from the channel clusters, averaged across time-points and frequencies within the respective cluster. Topographies depict grand-average GBA scalp distribution, averaged across time-points and frequencies of the corresponding effect. For all boxplots, whiskers indicate the interquartile range. The bold horizontal line indicates the distribution median. Dots indicate single-subject values. Black squares indicate the distribution mean. lTLR patients with apical resections (spared hippocampus, $n = 5$) are marked in blue. Brackets mark significant comparisons that specify interactive effects of *Group* and *Emotion* (***$p \le 0.001$). *Abbreviations*: GBA gamma-band activity, HC healthy controls, lTLR left temporal lobe resection, ns not significant, rTLR right temporal lobe resection.

statistically significant, were reduced in rTLR patients even beyond the P1, initial amygdalar feedback might also affect subsequent information processing although the role of downstream structures seems to increase. A similar pattern was found by another recent study on emotional picture processing following rTLR[35]. This could account for the reduced N1 emotion differentiation in rTLR as observed by Framorando and colleagues[19]. Of note, those authors found no emotion differentiation in the P1 in any group, which contrasts with our data. One possible explanation could be the variability of P1 emotion effects across studies that might be particularly limiting in small-n studies[4,47]. Generally, the largely preserved post-P1 emotion differentiation in the present rTLR patients is also consistent with the idea of an increasingly distributed network for threat processing over time[5,11]. Accordingly, we found an association of fusiform resection extent and emotion differentiation in the LPP (but not earlier components) in those patients. Additionally, for rTLR patients, the recognition bias for fearful relative to neutral faces was attenuated with larger LPP emotion differentiation in favor of fear. This might reflect a stronger reliance of rTLR patients on cognitive processes to support fear discrimination[34,48]. Future work should test such effects in more detail. Indeed, there is growing evidence emphasizing an alternative to the amygdala-centric view of emotion processing. This mechanism might be, in part, realized within the visual system itself[10].

Whereas rTLR had reduced ERP responses to fearful faces at very early processing stages, lTLR exhibited a larger fearful-neutral differentiation in ERP components from the N1 onward. Heightened fear processing after lTLR might explain previous reports of emotional dysregulation in patients with left temporal lobe strokes[49,50]. Hypervigilance has been found to be specific to damage to the bilateral basolateral amygdala in humans[51]. Overall, present data suggest that resection of specifically the left-hemispheric amygdala results in a disinhibition of fear-specific signaling and a resulting exaggeration of fear responses[52]. In fact, data might be explained through interhemispheric cooperation and the interplay of both amygdalae in normal emotion processing. Resting-state data by Tetereva and colleagues[53] demonstrate that the left amygdala exhibits the most prominent functional connectivity to the right amygdala, while the right amygdala exhibits higher interconnectivity than the left to sensory and associative areas. Gläscher and Adolphs (2003)[54] argue that the left amygdala decodes arousal signals from a stimulus, while the right amygdala activates the arousal system. Hence, the left mTL, potentially particularly the amygdala, seems to regulate cortical information processing, potentially by inhibiting disproportionate activation of the right amygdala and subsequent fear processing in the cortex. Alternatively, it is possible that lTLR-driven increases in processing of facial affect are caused by a dysfunctional feedback between the left middle temporal gyrus, relevant for emotion regulation[55,56], and the ipsilateral amygdala[57]. lTLR might also lead to re-organization processes that affect early visual perception as reflected in the topographically shifted P1 emotion differentiation. A similar topographic shift in the P1 was found for those patients during word processing[45]. Furthermore,

**Table 1 | Hemisphere-specific patterns of fear differentiation following TLR**

| Measure | Fear difference | B | t | p | d (fearful-neutral) | |
|---|---|---|---|---|---|---|
| P1 | rTLR ↓ | −0.081 | −2.138 | 0.025 | lTLR | 0.409 |
| | | | | | HC | 0.494 |
| | | | | | rTLR | −0.234 |
| N1 | lTLR ↑ | −0.081 | −2.103 | 0.040 | lTLR | −1.296 |
| | | | | | HC | −1.025 |
| | | | | | rTLR | −1.231 |
| EPN | lTLR ↑ | −0.080 | −1.741 | 0.084 | lTLR | −1.039 |
| | | | | | HC | −1.219 |
| | | | | | rTLR | −0.810 |
| LPP | lTLR ↑ | 0.094 | 2.441 | 0.017 | lTLR | 1.127 |
| | | | | | HC | 0.604 |
| | | | | | rTLR | 0.784 |
| GBA | rTLR ↓ | −0.019 | −3.687 | 0.0002 | lTLR | 0.115 |
| | | | | | HC | 0.236 |
| | | | | | rTLR | −0.856 |

*Note*. Beta coefficients are reported as unstandardized effect sizes. A downward/upward error indicates reduced/enhanced fear differentiation (higher amplitudes for fearful than neutral faces in P1, LPP and GBA, vice versa for the N1 and EPN) compared to the unweighted average across all groups. *Abbreviations: EPN* early posterior negativity, *HC* healthy controls, *GBA* gamma-band activity, *LPP* late positive potential, *lTLR* left temporal lobe resection, *rTLR* right temporal lobe resection.

adequate memory representations in lTLR patients seemed dissociated from visual salience[34,58] as shown by a missing association of EPN emotion differentiation with the memory bias. Of note, these mechanisms seemingly do not generalize to word processing[45]. Here, lTLR patients showed a rather consistent decrease in left-hemispheric emotion differentiation in the N1 and LPP components beyond the P1. Since word processing is thought to depend on lexical categorization in left fronto-temporal areas[59], cerebral networks for emotion differentiation in abstract and learned stimuli such as written words could differ from those recruited during the processing of biologically salient emotion cues like faces[60].

Turning to GBA, we propose that the right mTL is integral to a circuit-level computational process[26,27] that differentiates fearful and neutral expressions in right occipito-temporal areas. Here, rTLR patients over-represented neutral faces. Hence, we tentatively propose that a right-hemispheric and mTL-dependent GBA circuit is relevant for the differentiation of fearful and neutral faces through the relative de-prioritization of non-emotional faces. Hence, disrupted structural integrity of the right mTL might not affect the emotion-specific sparsification mechanism per se[10,26], but instead its competitive strength compared to the representation of neutral content. This hints towards potential qualitative differences in neural networks encoding fear and neutral content. Local prioritization of fearful faces in the right mTL might well depend on functional connections with the ipsilateral amygdala[61]. However, given the limited number of studies in this area, the functional network behind occipito-temporal GBA should be investigated in the future. Beyond the amygdala, it may rely on the structural integrity of the fusiform gyrus[31,62,63] or the temporal pole[64–67]. One future approach could be to investigate frequency coupling within and between the amygdala and visual cortices.

In contrast to the electrophysiological data, behavioral data revealed no effects of TLR on verbal report of subjective emotional experience, consistent with recent constructivist ideas suggesting that the conscious categorization of emotions does not necessarily link to one specific neural correlate[68]. Interestingly, fear differentiation in the N1 was associated with higher arousal scores for fearful than neutral faces across all groups. Hence, subjectively tangible emotional arousal might be first differentiated during

the N1 time-window[69]. This also supports the assumption that signals from the left amygdala determine the degree of stimulus arousal[54], given that changes in emotion differentiation in lTLR patients emerged specifically with the N1 time-window. Results from the recognition memory task do confirm an involvement of the right mTL, likely the right hippocampus[70], in memory for faces as rTLR patients performed more poorly. Moreover, all groups showed a recognition bias for fearful faces, which could stem from an adjustment of retrieval criteria in favor of fear-related stimuli by the frontal cortex[71].

Although these data advance our understanding of the neural mechanisms underlying emotional vision, their interpretation has some limits that need to be acknowledged. First, generalizability is limited given that our sample consisted of a clinical population. While we tried to control for between-sample confounds using a case-control study, it remains difficult to deduce the function of healthy brain tissue based on the comparison of a clinical sample with a healthy control group. Second, scalp EEG cannot reliably uncover activity within deeper brain regions[16]. Hence, to directly investigate amygdalar activity, intracranial recordings might be best suited for such analyses[39]. Similarly, the neural sources of the surface effects are still unknown, which future studies could approximate with inverse source modelling. Such analyses could also estimate the role of resection-induced changes in volume conductor properties. Relatedly, large cerebrospinal fluid cavities might have led to distortions of scalp topographies[72,73]. Additionally, although implicitly assumed, functional connectivity, e.g., between the amygdala and visual areas, was not tested in the present study, but could be analyzed with intracranial recordings or combined EEG and neuroimaging data in the future[74].

Overall, the present study provides compelling evidence for functionally distinct roles of the left and right the mTL during emotional vision. Previous models on emotional vision emphasized that the amygdala enhances the intensity of visual processing in favor of emotion[5]. Here, we demonstrate that this might adequately describe the contribution of the right amygdala to processes that generate early ERP components but not the complementary and potentially regulatory function of the left amygdala[25]. Importantly, to the best of our knowledge, robust data regarding the specific timing of those contributions were lacking. Furthermore, present data suggest that a comprehensive characterization of emotion processing necessitates the analysis of oscillatory synchronization within and potentially also between structures. These findings could inform clinical approaches to effectively regulate amygdala responses in anxiety and trauma disorders, as recently trialed by using neuromodulation[14,75]. Furthermore, they could have relevant implications for the surgical treatment of epilepsy in patients with comorbid psychiatric disorders. For instance, trauma symptoms were shown reduced after rTLR but heightened after lTLR[76,77]. Although these are only single cases and there is some conflicting data[78], these studies imply that interactions of unilateral TLR with psychiatric symptoms might be, at least in part, predicted based on resection side.

## Materials and methods
### Sample
Data from 18 lTLR patients (eight female), 18 rTLR patients (eight female), and 18 matched healthy controls (HC, nine female) with no past or acute neurological and psychiatric disorders are included in this report (Table 2). All groups were individually matched regarding age (+/− 1 year), sex, and education (highest educational qualification), resulting in a case-control study. Originally, 20 participants per group participated in this study. Two participants were excluded because of technical issues, three due to large perspiration and movement artifacts ($n = 3$), and one due to a large infarction in the left ventral visual pathway. All rTLR patients and HC were right-handed, four lTLR patients were left-handed. Two of those showed typical left-sided language lateralization. Language lateralization in the remaining two was unclear. Our sample sizes are considerably larger than those reported in previous publications[18,19] but we did not pre-determine sample sizes statistically. Nevertheless, data

**Table 2 | Sociodemographic and clinical details of the sample**

| | ITLR (*n* = 18) | | | rTLR (*n* = 18) | | | HC (*n* = 18) | | |
|---|---|---|---|---|---|---|---|---|---|
| | **Mean** | **SD** | **Range** | **Mean** | **SD** | **Range** | **Mean** | **SD** | **Range** |
| Age | 37.39 | 12.84 | 22–58 | 34.06 | 11.93 | 21–53 | 35.17 | 12.13 | 21–52 |
| Education (years) | 11.39 | 2.59 | 9–16 | 11.50 | 2.50 | 9–16 | 11.44 | 2.17 | 9–16 |
| STAI-S (state) | 35.89 | 10.86 | 21–62 | 40.28 | 9.96 | 27–62 | 32.89 | 5.29 | 25–41 |
| STAI-T (trait) | 41.06 | 12.07 | 24–67 | 40.28 | 10.78 | 24–63 | 34.61 | 8.93 | 24–58 |
| BDI | 7.76 | 7.98 | 0–34 | 7.78 | 6.03 | 0–24 | 3.05 | 3.03 | 0–10 |
| Months since resection | 55.44 | 29.62 | 24–101 | 48.06 | 20.44 | 24–81 | / | | |
| Age at onset (years) | 16.14 | 11.66 | 0.5–55 | 15.61 | 8.87 | 2–39 | / | | |
| Age at resection (years) | 32.72 | 12.06 | 16–56 | 30.06 | 11.36 | 17–50 | / | | |
| Resection volume (in voxels) | 12,941.94 | 3625.89 | 8564–21,973 | 15,376.44 | 4282.95 | 8904–25,560 | / | | |

*Note*. Both patient groups showed higher BDI scores than controls (ITLR versus HC: $t_{(20.322)}$ = 2.281, $p$ = 0.012; rTLR versus HC: $t_{(25.099)}$ = 2.965, $p$ = 0.030), but only two fulfilled the criteria for a moderate or severe depressive episode. In the STAI, both patient groups had marginally higher trait anxiety scores compared to the HC group (ITLR versus HC: $t_{(32.867)}$ = 1.716, $p$ = 0.096); rTLR versus HC: $t_{(31.334)}$ = 1.820, $p$ = 0.078). No group differences were found in the state scale ($F_{(1,81)}$ = 0.919, $p$ = 0.342). Patients did not differ in clinical variables (Months since resection: $t_{(30.199)}$ = −0.871, $p$ = 0.391; Age at epilepsy onset: $t_{(31.735)}$ = −0.153, $p$ = 0.880; Age at resection: $t_{(33.881)}$ = −0.683, $p$ = 0.500). Resection extent was smaller in ITLR patients compared to rTLR patients ($t_{(68.144)}$ = −2.603, $p$ = 0.011).
*Abbreviations*: *BDI* Beck Depression Inventory[103], *HC* healthy controls, *ITLR* left temporal lobe resection, *rTLR* right temporal lobe resection, *STAI-S* State-Trait Anxiety Inventory, State subscale, *STAI-T* State-Trait Anxiety Inventory, Trait subscale[104].

simulations indicate that the present participant and trial numbers should be sufficient to investigate the measures of interest[79].

For all patients, temporal lobe resections included the amygdala and surrounding tissue as well as the hippocampus (anteromedial temporal lobe resection). In five lTLR patients, the hippocampus was spared (apical resection). Figure 1 shows the resection extents and their overlap. Patients were surgically treated at the Department of Epileptology (Krankenhaus Mara) of Bielefeld University for pharmacoresistant temporal lobe epilepsy. Testing was done at least 24 months after surgery to ensure sufficient post-operative adjustment in all patients. No patients had amygdala lesions prior to the surgery, as identified from structural imaging and post-operative histology. Patients' surgery outcome was classified according to criteria proposed by Engel[80]. Thirty patients were free of disabling seizures (Engel I), two patients reported rare disabling seizures (Engel II), and four had less favorable outcomes (Engel III & IV). Eighteen patients (9 rTLR, 9 lTLR) confirmed to use anti-seizure medication at the time of testing. Participants gave written informed consent according to the Declaration of Helsinki and received a monetary reward of 100 Euros for their participation in the entire study, which consisted of EEG, fMRI, and extensive neuropsychological testing. fMRI and neuropsychology data are reported elsewhere[8,81]. The study was approved by the ethics committee of the German Psychological Association (DGPs). All ethical regulations relevant to human research participants were followed.

**Stimuli**

Images of faces were derived from the FACES database of the Max Planck Society for the Advancement of Science with prior permission[82], the NimStim Set of Facial Expressions[83], and the Karolinska Directed Emotional Faces database[84]. Only heads and shoulders, in color, were placed centrally on a white background and scaled to a uniform width and height (430 pixels, 9.28° vertical visual angle [centered]).

For each participant, from a set of 120 identities that were balanced for sex and age, 40 identities per facial expression (fearful, neutral) were pseudo-randomly chosen and presented in a passive viewing paradigm. Additionally, 20 identities per expression were shown as novel faces in a recognition experiment 24 h after incidental encoding. Face selection did not vary between groups. Additionally, six happy expressions were presented randomly across the block to reduce predictability of subsequent stimuli.

**Procedure**

Faces were presented centrally on a 17-inch TFT monitor with a refresh rate of 60 Hz that was positioned on a table in front of the participants. Participants were seated in a comfortable chair at about 70 cm from the center of the screen. They were instructed to attentively view randomly presented negative and neutral faces. This task was chosen to ensure implicit emotion processing without distracting task effects[85]. The selected stimulus set of 80 faces was divided into four blocks and presented in randomized order. Participants were instructed to remain as quiet as possible and refrain from eye and body movements during each stimulation block. Pseudo-randomly alternating with the blocked presentation of faces were same-length blocks of visual scenes[35] and word stimuli[45] to reduce habituation to a specific stimulus type. Self-paced breaks were offered between stimulation blocks. After a run of four blocks, the entire presentation was repeated to facilitate incidental encoding, and every face was presented twice, leading to 344 trials. Stimuli were shown for 800 ms. A blank black screen served as the inter-stimulus interval which was presented for 1500 ms ( ± 200 ms jitter). Stimulus presentation was controlled using the Presentation® Software (Version 18.1, Neurobehavioral Systems Inc., Berkeley, CA, http://www.neurobs.com) running on a DELL OptiPlex 755 PC. EEG testing took about 40 min. Twenty-four hours after the stimulus presentation, participants were presented again with 80 identities with fearful or neutral expressions out of which 20 identities per expression were already presented in the main EEG experiment. Participants were then asked to classify the faces as either "old" or "new" with a pseudo-randomly assigned button press (left and right arrow). Stimuli were shown until response or for a maximum of 2000 ms. After 2000 ms, a black screen with a central white question mark instructed the participants to respond. It was shown until a response or a maximum of 1000 ms. A black screen with a central white fixation cross served as the inter-stimulus interval and was presented for 1500 ms. Finally, participants rated both valence and arousal on a 7-point Likert scale of 10 fearful and 10 neutral faces that were randomly chosen from the pool of presented faces. Because the electrophysiological data were the main measures of interest, full data and analysis of recognition and rating tasks are reported in Supplementary Tables 1–4 and Supplementary Fig. 1).

**Resection mapping**

To evaluate resection extent, structural MRI images were used. MRI data were recorded using a 3T Magnetom Verio Scanner (Siemens, Erlangen, Germany) with a 12-channel head coil. High-resolution

T1-weighed structural images were acquired in 192 sagittal slices (TR = 1900 ms, TE = 2.5 ms, voxel size = 0.75 × 0.75 × 0.8 mm, matrix size = 320 × 320 × 192). Resected areas were traced manually in the individual structural T1 images of the patients and converted to Montreal Neurological Institute (MNI) space by applying the deformation fields derived from normalizing procedure in the pre-processing of the respective patient to the resection mask.

## EEG processing
During stimulus incidental encoding, the EEG was continuously recorded from 128 active BioSemi Ag-AgCl electrodes using the ActiView software (http://www.biosemi.com). The horizontal electrooculogram (EOG) was measured with two electrodes placed on the outer canthi of the left and right eyes. Two further electrodes for the vertical EOG were placed over and under the left eye. The sampling rate was set to 1024 Hz. Impedances were kept below 25 kΩ, in line with system recommendations. Online recording was referenced to the Cz electrode. Pre-processing of offline data was per-formed using the FieldTrip toolbox[86] (version 20200121) in MATLAB (version 2019b, The MathWorks Inc.). Offline, data was re-referenced to the common average. A high-pass Butterworth first-order zero-phase filter of 0.1 Hz, a 49-51 Hz discrete Fourier transform line noise filter, and a low-pass first-order zero-phase Butterworth filter of 200 Hz were applied. Filtered data were cut into segments of −1500 to 2100 ms relative to stimulus onset and then down-sampled to 500 Hz. Omission of oculomotor and technical (i.e., line or channel noise) artefacts was done by omitting respective inde-pendent components. Finally, if single trials still exceeded a maximum amplitude range of 250 μV, those were excluded as well. On average, 16.11% of trials per condition were rejected. The rejection rate did not differ between conditions ($t_{(53)} = 0.084$, $p = 0.933$), but marginally fewer trials were rejected in the rTLR data compared to HC ($t_{(35)} = 2.300$, $p = 0.083$) and lTLR ($t_{(35)} = 2.260$, $p = 0.083$). On average, 1.92% of electrodes were interpolated per participant. The channel interpolation did not differ between groups (all $p$s ≥ 0.222).

## ERP analysis
To ensure methodological comparability between TLR studies[18,19], we analyzed four ERP components with pre-defined temporal and spatial occurrences. These allow inferences about specific levels of sensory and higher cognitive processing of emotional value. For this analysis, an addi-tional low-pass first-order zero-phase Butterworth filter of 40 Hz was applied to ensure accurate detection of sharp peak components[87]. Per channel and participant, the ERP was extracted from −200 to 800 ms relative to stimulus onset. Baseline-correction used 200 ms before stimulus onset. Data were then averaged across trials for each participant. Electrode clusters for each component were selected in line with the literature. To investigate the P1, a time-window ranging from 90 to 130 ms was segmented using two symmetrical clusters of five occipitoparietal electrodes each (Fig. 2a[88,89]). Two symmetrical channel clusters of eight occipitoparietal electrodes were used to analyze the N1 (140–200 ms, Fig. 2b[19,35]) and EPN (200 to 350 ms, Fig. 2c[34,35]). For the LPP, data were extracted from 400 to 800 ms at two centroparietal clusters comprising 13 channels each (Fig. 2d[34,35]). The midline electrodes were equally allocated to the right and left LPP cluster. ERP component extraction was performed using custom code and the FieldTrip toolbox[86] (version 20200121) in MATLAB (version 2019b, The Math Works Inc.).

Additionally, post-hoc assessment of the differential P1 topography (fearful-neutral) showed a topographic shift of the emotion differentiation in lTLR patients. Therefore, we conducted a separate analysis of P1 amplitudes in a shifted electrode cluster (BioSemi channel labels: D24, D25, D31, D32, B10, B11, B14, B15, see Supplementary Fig. 2).

## Spectral analysis
From −1500 ms to 2100 ms relative to stimulus onset, the time-frequency power spectrum of each single trial was calculated at every scalp channel from 30 to 90 Hz. Data were down-sampled to 200 Hz prior to analysis. We used Slepian tapers with a fixed window length of 200 ms and applied spectral smoothing through multi-tapering (three tapers) of 20 Hz for all frequencies[40]. This way, an equal number of time-frequency bins across the time-frequency spectrum was obtained for statistical analysis[90]. A baseline correction for the frequency calculations was applied by expressing the gamma-power as a relative change to the power in the baseline of −600 to −100 ms pre-stimulus interval. Finally, epochs containing residual artefacts (e.g., muscular activity, technical noise) that were not previously detected in the time domain were excluded from analysis after careful trial-by-trial visual inspection. This rejection was conducted without knowledge of the condition that a trial belonged to. However, it was not blind regarding the group allocation. Nonetheless, this did not lead to systematic differences in trial counts between groups (see paragraph "EEG processing"). Spectral analysis was performed using the custom code and the FieldTrip toolbox[86] (version 20200121) in MATLAB (version 2019b, The MathWorks Inc.).

## Statistics and reproducibility
**General approach.** For all statistical analyses, we evaluated mixed-effects linear regression models in R[91] running in RStudio (version 2023.03.1)[92]. For electrophysiological data, we used single-trial data to account for individual trial-by-trial variability and to elegantly handle the hierarchical structure of the mixed-design data[91]. Factors were effect-coded as repeated a priori contrasts that compare the means of factor levels to the unweighted average[93] within a linear mixed model (LMMs, lme4 v.1.1-34). This method allows direct inferences about the direction of a given effect and interaction due to a priori defined contrasts while simultaneously accounting for the variance of all factor levels. Models were fitted with by-ERP component random slopes and by-subject random intercepts to account for repeated measures and trial dependencies[94]. The significance of model coefficients was tested and reported with two-sided Satterthwaite $t$-tests. Unstandardized fixed effects coefficients are reported as effect sizes in the LMMs. Significance of beta-coefficient $t$-values was tested with Montecarlo permutation tests. The alpha-level was set to 0.05. Data was randomly shuffled between conditions (single trials of one condition were kept together while permuting conditions), and after each permutation step, an LMM was calculated. Permutation steps were repeated 1000 times. Empirical $t$-values were then compared to the distribution of permuted $t$-values[95]. The permutation distribution is a data-driven non-para-metric distribution. Therefore, no degrees of freedom are given. Post-hoc comparisons were performed with the predicted marginal means using emmeans[96] (v.1.8.8). With Holm corrections of the post-hoc tests, we accounted for the inflation of false positives in multiple testing per measure[97]. For the post-hoc tests, Cohen's $d$ is reported as the effect size. It is interpreted as small when $d = 0.20$, medium when $d = 0.50$, and large when $d = 0.80$[98].

**Behavioral data.** Stimulus ratings, both valence and arousal, were cal-culated as mean ratings from a discrete 7-point Likert scale across all stimuli of the same emotion condition per subject. To determine recognition accuracy while controlling for a response bias, the corrected discrimination index (DI: Hits—false alarms) was reported according to the two-high threshold model[99]. The response bias (Bias) was calculated as false alarms divided by 1-recognition accuracy.

For all behavioral data, we averaged data across trials for each participant and condition to calculate the indices of interest. The models included fixed effects for the experimental factors Group (HC, lTLR, rTLR) and Emotion (fearful, neutral) and their interaction as follows:

$$Data \sim Group + Emotion + Group * Emotion$$

Random effects were omitted for the analysis of average behavioral data to avoid overparameterization. Each measure was analyzed with its own statistical model.

**ERP components**. All ERP components were included in one LMM. The models included the fixed effects for the experimental factors *ERP component* (P1, N1, and LPP), *Group* (HC, lTLR, rTLR), *Emotion* (fearful, neutral), and *Hemisphere* (left, right) and their interaction as follows:

$$Data \sim ERP + Group + Emotion + Hemisphere$$
$$+ ERP\ component * Group * Emotion * Hemisphere$$
$$+ (1 + ERP\ component|Subject)$$

The contrasts compare the mean of one specific factor combination to the unweighted average across all factor combinations, facilitating a direct interpretation of the effect size in data units while accounting for all factor levels and data dependencies. Contrasts can be interpreted as follows: the beta coefficient indicates the difference in units for the specific factor level/combination compared to the unweighted average across all levels/combinations. Dependency between trials and ERP component amplitudes was accounted for by the specification of the random effects. Because of the sum-to-zero constraint in effect-coded models, no contrasts tested for effects in the EPN and HC group. A separate model that investigates interactions in the EPN is documented in Supplementary Table 7. Estimated marginal means and post hoc tests of all groups and ERP components can be found in Supplementary Table 6.

**GBA**. Because, to the best of our knowledge, no study investigated effects of TLR on emotion differentiation in GBA, we opted for a data-driven approach in the spectral analysis. We aimed to identify any time-frequency points of significant effects with respect to GBA power. For this, GBA was analyzed from 35 to 90 Hz[40] across all scalp channels from 0 to 800 ms, corresponding to the analyzed time-window for the ERPs. We specified a model with the fixed effects for the experimental factors *Group* (HC, lTLR, rTLR) and *Emotion* (fearful, neutral) as follows:

$$Data \sim Group + Emotion + Group * Emotion + (1|Subject)$$

To assess pointwise significance of LMM coefficients while effectively accounting for heightened false positives in multiple comparisons of multidimensional data, we utilized cluster-based corrections of Montecarlo permutation tests. Clustering was performed using the maximum summation approach[100]. The analysis code was modified based on the analysis by Méndez-Bértolo and colleagues[6]. *T*-values of each cluster were summed and the greatest sum among all clusters was entered into the permutation distribution. Significant clusters were determined based on both temporal and frequency adjacency with a cluster threshold of $\alpha = 0.01$, correcting for multiple comparisons on the cluster-level. Only clusters that span more than 10 ms and 5 Hz are reported. Estimated marginal means and post-hoc tests of all groups in clusters indicating an interaction can be found in Supplementary Table 8.

**Behavior—electrophysiology associations**. We investigated whether emotion differentiation in electrophysiological data was associated with systematic changes in emotion differentiation in the stimulus ratings and recognition performance. To do so, we calculated the difference between responses to fearful versus neutral faces for both behavioral and electrophysiological measures. For each behavioral measure (arousal, valence recognition performance, memory bias), a linear mixed model (LMM) was calculated, testing for the effects of the differential ERP component amplitude or GBA power, group, and their interaction on behavioral emotion differentiation. Models were fitted with by-group random intercepts. Full model coefficients are detailed in Supplementary Table 9. Significant brain-behavior associations are illustrated in Supplementary Fig. 4.

**Control analyses**. To explore whether any effect of emotion might be moderated by clinical factors such as the age at resection, age at epilepsy onset, time since resection, and BDI scores, we calculated the fear-neutral

difference and entered them as dependent variables into our control model. Beforementioned clinical variables served as predictors. Furthermore, to test whether the extent of hippocampal[101] and fusiform gyrus[63] (FG) resections might have contributed to altered electrophysiological emotion differentiation in patients, we investigated whether the magnitude of respective resections associated with changes in differential emotion processing in the EEG measures. Resection volume was quantified by assessing the overlap between individual lesion masks and a standardized anatomical atlas. Manually constructed lesions masks (see paragraph "resection mapping") were first resampled to match the dimensions of the Automated Anatomical Labeling[102] (AAL) atlas using nearest-neighbor interpolation. A binary lesion mask was then generated to include all nonzero voxels, representing the extent of resection. To determine the resection volume of specific brain regions, the resampled lesion mask was overlaid onto the atlas, and the number of voxels corresponding to the respective region was extracted. The proportion of region-specific voxels affected by resection was then calculated relative to the total number of region-specific voxels in the atlas. All image processing and quantifications were performed using nibabel and nilearn in Python 3.

Predictors were centered to avoid multicollinearity. Models were individually calculated for lTLR and rTLR patients to avoid over-parameterization and to ensure a more detailed description of each patient sample. This approach was identical for the ERP components and GBA. For the GBA data, we modelled data within the *Group* x *Emotion* interaction clusters identified through the cluster-based LMM approach. Full model coefficients of ERP data are documented in Supplementary Table 10. Full model coefficients of GBA data are documented in Supplementary Table 11.

## Reporting summary

Further information on research design is available in the Nature Portfolio Reporting Summary linked to this article.

## Data availability

All data needed to evaluate the conclusions are present in the paper or the supporting information. The datasets generated in this study are available under this link: https://gitlab.ub.uni-bielefeld.de/ae02/weidnertlr so are numerical source data for graphs and charts. Raw data are available from the authors upon reasonable request to the corresponding author.

## Code availability

The codes generated in this study are available under this link: https://gitlab.ub.uni-bielefeld.de/ae02/weidnertlr.

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

## Acknowledgements

This work was funded by grants from the German Research Foundation: Deutsche Forschungsgemeinschaft, Grant/Award Numbers BI1254/8-1 and KI1286/6-1. We thank Prof. Dr. Stephan Moratti, Dr. Alba Peris-Yagüe, and Dr. Manuela Costa for helping with the analysis of time-frequency data. Thank you to Dr. Sebastian Geukes for helping with the LMM pipeline. Thank you to Prof. Dr. Axel Mayer for reviewing the statistical analysis. We would also like to express our gratitude to all the participants.

## Author contributions

E.W. conducted participant testing and performed data processing, analysis, and visualization. M.M. and L.M.R. conducted participant recruitment and testing. L.M.R. performed structural MRI scans and processing of MRI data. C.G.B. contributed to the study design and supported patient recruitment and organization of participant testing in the hospital (Krankenhaus Mara, Bielefeld). J.K. contributed to the study design and supervised data analysis. E.W. drafted and revised the manuscript under supervision of J.K. L.M.R., M.M., and C.G.B. helped to draft and revise the manuscript. All authors read and approved of the final manuscript.

## Funding

## Competing interests

The authors declare no competing interests.
