## [Transparent Peer Review file · Communications Biology]

Fear perception as a function of hemisphere- and time-specific dynamics in the medial temporal lobes

Corresponding Author: Dr Enya Weidner

Version 0:

Reviewer comments:

Reviewer #1

(Remarks to the Author)

Fear perception as a function of hemisphere- and time-specific dynamics in the medial temporal lobes

[Paper # COMMSBIO-25-0793]

Enya M. Weidner, Lea Marie Reisch, Malena Mielke, Christian G. Bien & Johanna Kissler

The medial temporal lobes and particularly the amygdalae appear to enhance the visual processing of fearful faces. The exact mechanisms underlying this complex dynamic are still relatively unclear. Using high-density 128 channel EEG, this study compared event-related potentials (ERPs) and induced gamma-band activity (GBA) from 18 patients with left- and 18 patients with right hemispheric temporal lobe resections (ITLR and rTLR) and 18 matched healthy controls when viewing randomly presented fearful and neutral facial expressions. Results revealed that the rTLR patients lacked fear-neutral differentiation in the P1 component and showed stronger gamma-band activity (GBA) for neutral compared to fearful faces over right hemispheric (i.e. ipsi-resectional) occipito-temporal areas.

By contrast, ITLR patients showed stronger emotion differentiation in the later N1, EPN and LPP components. The authors interpret this finding to mean that the right mTL, possibly the amygdala in particular, supports rapid shifts in attention towards fear and appears to coordinate fear-neutral differentiation, which is reflected in particular in the GBA.

In comparison to previous related studies, this study is based on quite large and well balanced datasets. The analysis is state of the art and results and discussion are presented in an excellent and very understandable way. The results expand our understanding of the obviously hemisphere specific roles of the medial temporal lobes for emotional processes and thus, are of great importance to the clinical and scientific community.

There are however some minor comments in the following which should be addressed before publication:

Minor Comments:

Abstract:

"These results constrain current models of emotional vision".

Too general, please specify. Probably "extend" instead of "constrain"?

Whole MS:

The manuscript often refers to ERPs, while specific ERP components are meant. For instance: "... amplitude values of four ERPs that are typically sensitive to visual and higher cognitive emotion processing".

This should be adapted throughout the manuscript.

Intro:

"Building upon previous research, we expected at least one patient group to exhibit reduced differentiation of fearful from neutral faces".

This hypothesis seems too unspecific. It would be good to explain again in one sentence why possibly only one of the two patient groups could show a reduced differentiation of anxious and neutral faces and which of the resection groups should

show the least differentiation.

Results:

"For ITLR patients, P1 emotion differentiation was present in a forward-shifted electrode cluster (ITLR: $\beta_{\text{fearful}} - \text{neutral} = 0.016$, $p = .050$, $d = 0.410$; BioSemi channel labels: D24, D25, D31, D32, B10, B11, B14, B15)".

It is unclear how this electrode cluster was included in the analysis. Is this a post-hoc analysis or was this cluster chosen instead of the left hemispheric a priori defined sensor group?

Figure 2: Group labels are missing for the fearful minus neutral difference plots.

Next to the star in Figure 2A fearful minus neutral difference plots, a 1 has probably been inserted by mistake.

"A main effect of emotion was found over left occipital contacts from 85 – 195 ms in a frequency range of 70-85 Hz, driven by a more right-lateralized GBA increase in response to fearful than neutral faces (Fig. 4)".

Where is this left sided main effect of emotion visible in figure 4? Where is the right lateralized GBA increase visible? Why is the one driving by the other?

Even if there are no differences between the groups in terms of arousal and valence, it would be interesting to investigate whether there is a connection between the behavioral and electrophysiological data. The same applies to the recognition scores: for example, do the subjects in the rTLR group with the poorest recognition also show the strongest gamma band differentiation?

Discussion:

Important limitations of this study are missing:

Even though 18 subjects per group are significantly more than were examined in related previous studies and recruitment of such specific patients is difficult, it should still be noted that these are still relatively small groups for EEG examination and the results should therefore be interpreted with caution.

Second, the a-priori defined channel clusters for analyzing the ERP components were based on studies of healthy control subjects. The resection-induced changes in volume conductor properties within the resection area might have led to shifts of ERP topographies. The contribution of these volume conductor related changes on ERP topographies is rather unclear but could be estimated on the basis of highly realistic head models and inverse source modeling.

Even if the restrictions due to the maximum length of the manuscript in this journal are quite strict, it would be extremely helpful if the main results of this study were placed in the context of the corresponding studies on emotional word processing (76) and the processing of emotional scenes (33), as these studies are based on predominantly overlapping subject groups. Perhaps the most important cross-stimulus results and the most important stimulus-specific differences between the studies could be reproduced here in the discussion. A more detailed discussion of this could be provided in the supplementary material.

Methods:

"Finally, epochs containing residual artefacts (e.g., muscular activity, technical noise) that were not previously detected in the time domain were excluded from analysis after careful trial-by-trial visual inspection".

Was this artefact rejection conducted without knowledge of group membership (blinded)? If not, this should be discussed here as a limitation.

Reviewer #2

(Remarks to the Author)

The authors investigated hemispheric differences in the contributions of medial temporal lobe (mTL) to fearful face processing. They compared patients with left mTL resection and right mTL resection with controls (18 participants per group). Behaviorally, no differences among groups were seen for emotion effects on face memory recognition (but generally reduced memory for both neutral and fearful faces in right TLR patients). A P1 amplitude increase for fearful versus neutral faces that was normally observed in controls (90-130 ms post-stimulus) was attenuated in rightTLR patients. Later ERP increases for fearful vs. neutral faces (N1, EPN, LPP) were abnormally amplified in ITLR patients, relative to controls, interpreted as left mTL failing to regulate fear information transmission. Finally, gamma response to neutral faces was also abnormally heightened at 95-300 ms in rTLR patients vs controls.

The study presents highly relevant data, and the results may provide important insights to human affective neuroscience. I

have previously reviewed this manuscript for another journal and the authors have addressed most of my previous concerns to full satisfaction. The methodology used seems appropriate and is described in sufficient detail to allow for reproducibility. However, I still have the following comments.

-«Resection extent was smaller in ITLR patients compared to rTLR patients». Although this did not have an overall impact on the results, it seems like patients' lesions could extend into ventral temporal cortex (Fig. 1), affecting fusiform or inferior temporal gyrus. Was temporal visual cortex differentially affected for rTLR vs ITLR patients, potentially explaining that rTLR patients showed particularly weak P1 effects for fearful faces? This is important, as the main claim is that the right amygdala is not sending inputs to the visual cortex in rTLR patients.

-Additionally, resection encompassed complete removal of amygdala in all patients, but 5 left TLR patients had spared hippocampus. Could this also have any impact on the EEG results across groups? Did patients with spared hippocampus show a different pattern relative to the rest of their group? Emotionally salient stimuli engage amygdala-hippocampus interactions (e.g. Zheng et al., NatCommunications; 2017). This could also have had a differential impact on habituation to faces, and thus on the resulting EEG results. Again, there is no overall effect of "resection extent" on the results, but a more specific reference to the hippocampal resection, or lack thereof, could be provided.

-P1: one of the main claims is that the earlier (P1) emotion response is attenuated in rTLR but unaffected for ITLR patients. However, tables S5 and S6 (and Fig. 3) are confusing in this regard, as they show no fear effect in ITLR patients either. Instead, the P1 emotion effect seen in ITLR patients is reportedly in a slightly different location: «For ITLR patients, P1 emotion differentiation was present in a forward-shifted electrode cluster (ITLR: $\beta_{\text{fearful}} - \text{neutral} = 0.016$, $p = .050$, $d = 0.410$; BioSemi channel labels: D24, D25, D31, D32, B10, B11, B14, B15). This should be better specified in Fig. 3 (left panel). It should also be specified (if that is the case) that no such effects were observed in this cluster for rTLR patients. Also, why was it shifted in ITLR patients?

Minor:

-In Table 2, it is said: «In the STAI, both patient groups had marginally lower trait anxiety scores compared to the HC group» but the table shows the opposite pattern (higher STAI trait scores for patients vs. healthy controls).

-Please revise p-values, as one p-value in Table S6 is negative, and some p-values reported in the main text (e.g. page 6) do not match those reported in Table S5.

Version 1:

Reviewer comments:

Reviewer #1

(Remarks to the Author)

The authors have discussed and dealt with all comments and questions to my complete satisfaction. I would like to congratulate you on this excellent manuscript. I would recommend that this paper be accepted in its present form.

Reviewer #2

(Remarks to the Author)

I thank the authors for thoroughly addressing all my remaining concerns. The manuscript has improved considerably and I have no further comments.

MAJOR REVISION MANUSCRIPT COMMSBIO-25-0793

Reviewers' 1 comments:

We are very grateful for your helpful and constructive feedback. We feel that your comments have much improved the quality of the manuscript. Thank you for taking the time to read and review our manuscript. Below, you will find a point-by-point response to the concerns raised. Responses are highlighted in blue text. Major insertions to the manuscript are highlighted in yellow. We have also tried to make the writing more accessible in places and corrected some typographical errors without highlighting each minor change in wording. The analysis scripts on GitLab are updated according to the latest modifications in response to your requests.

Minor Comments:

Abstract: "These results constrain current models of emotional vision". Too general, please specify. Probably "extend" instead of "constrain"?

Thank you for this suggestion. We have further elaborated this point in the abstract (l. 33):

"These results reveal complementary, lateralized, and time-specific roles of the medial temporal lobes in fear processing, thereby refining models of emotional vision."

Whole MS: The manuscript often refers to ERPs, while specific ERP components are meant. For instance: "... amplitude values of four ERPs that are typically sensitive to visual and higher cognitive emotion processing". This should be adapted throughout the manuscript.

We agree that this is an important detail and changed the wording throughout the manuscript, e.g., line 82:

*"Therefore, the present study explores the time and laterality-specific contributions of the mTL to the emotion-sensitive ERP **components**"*

Intro:

"Building upon previous research, we expected at least one patient group to exhibit reduced differentiation of fearful from neutral faces". This hypothesis seems too unspecific. It would be good to explain again in one sentence why possibly only one of the two patient groups could show a reduced differentiation of anxious and neutral faces and which of the resection groups should show the least differentiation.

Thank you for pointing this out. Given the sparse and partly contradictory data in this field, it is actually not trivial to deduce specific hypotheses. Still, based on previous data by Framorando and colleagues (2021), we could expect at least rTLR patients to differ from controls (attenuated emotion processing in earlier components) and we added this to the hypotheses. Since there is, to the best of our knowledge, no data on the effects of ITLR on facial emotion differentiation, we only specified tentative hypotheses for those patients. We have changed the hypothesis paragraph as follows (l. 87 onward):

"Building on previous research,¹⁸ we expected that rTLR patients would exhibit reduced emotion differentiation in early ERP components. Given the sparse data on emotion differentiation in ITLR patients, we did not formulate directional a priori hypotheses for this group. Nevertheless, based on previous findings of altered fear processing in individuals with left-hemispheric amygdala lesions,¹⁷ we anticipated that their responses would deviate from the overall group mean."

Results:

"For ITLR patients, P1 emotion differentiation was present in a forward-shifted electrode cluster (ITLR: $\beta_{\text{fearful} - \text{neutral}} = 0.016$, $p = .050$, $d = 0.410$; BioSemi channel labels: D24, D25, D31, D32, B10, B11, B14, B15)". It is unclear how this electrode cluster was included in the analysis. Is this a post-hoc analysis or was this cluster chosen instead of the left hemispheric a priori defined sensor group?

This was indeed a post-hoc analysis that we conducted after inspecting the difference topography (fearful-neutral) in the P1. For transparency, we clarified this in the methods section (l. 451 onward):

"Additionally, post-hoc assessment of the differential P1 topography (fearful-neutral) showed a topographic shift of the emotion differentiation in ITLR patients. Therefore, we conducted a separate analysis of P1 amplitudes in a shifted electrode cluster (BioSemi channel labels: D24, D25, D31, D32, B10, B11, B14, B15, see Supplementary Fig. 2)."

Furthermore, coefficients for all groups from this electrode cluster are now included in Supplementary Table 6. We also added the distribution of single-subject P1 amplitude values from this cluster as a separate boxplot in Supplementary Fig. 2.

We again emphasize that this was a post-hoc analysis in the results section (l. 127 onward):

"For ITLR patients, higher P1 amplitudes in response to fearful than neutral faces were present in a forward-shifted electrode cluster (ITLR: $\beta_{\text{fearful} - \text{neutral}} = 0.106$, $p = .050$, $d = 0.410$, Supplementary Fig. 2) but not in the a priori defined cluster ($\beta_{\text{fearful} - \text{neutral}} = 0-119$, $p = .277$, $d = 0.185$). The shifted cluster was chosen based on post-hoc assessment of the differential (fearful-neutral) P1 topography in ITLR patients. No other group showed significant emotion differentiation in the shifted cluster (HC: $\beta_{\text{fearful} - \text{neutral}} = 0.062$, $p = .270$, $d = 0.334$, rTLR: $\beta_{\text{fearful} - \text{neutral}} = -0.023$, $p = .696$, $d = -0.104$ (Supplementary Table 6))."

Figure 2: Group labels are missing for the fearful minus neutral difference plots. Next to the star in Figure 2A fearful minus neutral difference plots, a 1 has probably been inserted by mistake.

Thanks! Apologies for the oversight. The groups have now been added to the plots. The "1" was intended to refer to a footnote to show that the effect in ITLR patients was conditional (it depended on the shifted electrode cluster) but we agree that the way we depicted it was unnecessarily confusing. We now added a note to the caption itself (l. 169):

"Since significant emotion differentiation in ITLR patients was shifted forwards ($\beta_{\text{fearful} - \text{neutral}} = 0.106$, $p = .050$, $d = 0.410$; Supplementary Fig. 2) and not found in the a priori defined cluster ($\beta_{\text{fearful} - \text{neutral}} = 0-119$, $p = .277$, $d = 0.185$), the significance marking is depicted in brackets. "

"A main effect of emotion was found over left occipital contacts from 85 – 195 ms in a frequency range of 70-85 Hz, driven by a more right-lateralized GBA increase in response to fearful than neutral faces (Fig. 4)". Where is this left sided main effect of emotion visible in figure 4? Where is the right lateralized GBA increase visible? Why is the one driving by the other?

Thank you, we agree that this was potentially misleading. We worded this differently to depict the effect more clearly (l.177 onward):

"A main effect of Emotion was found over left occipital contacts from 85 – 195 ms in a frequency range of 70-85 Hz, driven by higher GBA power in response to neutral than fearful faces."

We now include plots that isolate the main effects in the Supplementary Fig. 3 In this regard, we also decided to only report effects with a cluster size larger than 10 ms and 5 Hz since smaller clusters are difficult to interpret. We state this in the methods section accordingly.

Even if there are no differences between the groups in terms of arousal and valence, it would be interesting to investigate whether there is a connection between the behavioral and electrophysiological data. The same applies to the recognition scores: for example, do the subjects in the rTLR group with the poorest recognition also show the strongest gamma band differentiation?

This is a very interesting analysis, thank you for this suggestion. We had initially omitted these analyses for brevity but are happy to provide them now. Our report now includes additional analyses in the results section and Supplement to investigate whether differential ERP amplitude/gamma power predict differential behavioral performance and rating scores. The approach is described in detail in the methods section (l. 534 onward):

“We investigated whether emotion differentiation in electrophysiological data was associated with systematic changes in emotion differentiation in the stimulus ratings and recognition performance. To do so, we calculated the difference between responses to fearful versus neutral faces for both behavioral and electrophysiological measures. For each behavioral measure (arousal, valence recognition performance, memory bias), a linear mixed model (LMM) was calculated, testing for the effects of the differential ERP component amplitude or GBA power, group, and their interaction on behavioral emotion differentiation. Models were fitted with by-group random intercepts. Full model coefficients are detailed in Supplementary Table 9. Significant brain-behavior associations are illustrated in Supplementary Fig. 4.”

Interestingly, we did find a positive association between EPN and LPP emotion differentiation and the recognition bias in some groups (l. 143 onward):

“For the HC and rTLR group, a larger difference of EPN amplitudes in favor of fearful faces was indicative of an enhanced memory bias for fearful relative to neutral faces (HC: $\beta_{\text{fearful-neutral}} = -0.177$, $t_{(35)} = -1.852$, $p = .072$, rTLR: $\beta_{\text{fearful-neutral}} = -0.161$, $t_{(35)} = -2.099$, $p = .043$). This was not present in lTLR patients ($\beta_{\text{fearful-neutral}} = 0.103$, $t_{(35)} = 1.321$, $p = .195$), as also reflected in the Group \times EPN interaction ($\beta_{\text{Group (rTLR)} \times \text{EPN}} = 0.182$, $p = .011$; Supplementary Fig.4, Supplementary Table 9). Furthermore, only for rTLR patients was there an association between LPP emotion differentiation and the differential recognition bias: When the difference in LPP amplitudes between fearful and neutral faces decreased, the recognition bias increased for fearful relative to neutral faces ($\beta_{\text{Group (rTLR)} \times \text{LPP}} = -0.201$, $p = .030$; Supplementary Fig. 4, Supplementary Table 9; rTLR: $\beta_{\text{fearful-neutral}} = -0.267$, $t_{(35)} = -2.333$, $p = .026$, HC: $\beta_{\text{fearful-neutral}} = 0.061$, $t_{(35)} = 0.880$, $p = .385$, lTLR: $\beta_{\text{fearful-neutral}} = 0.008$, $t_{(35)} = 0.102$, $p = .920$). Additionally, the analyses revealed a significant association between N1 emotion differentiation and arousal ratings across all groups. The higher the N1 amplitude difference in favor of fearful faces, the higher the difference in arousal ratings between fearful and neutral faces ($\beta_{\text{N1}} = -0.874$, $p = .008$). No significant effects were found for the valence ratings (all $ps > .100$).”

Since this potentially reveals how some patient groups (over)compensate for the resection, we added the following to the discussion (l. 251 onward):

“Generally, the largely preserved post-P1 emotion differentiation in the present rTLR patients is also consistent with the idea of an increasingly distributed network for threat processing over time.^{5,11} Accordingly, we found an association of fusiform resection extent and emotion differentiation in the LPP (but not earlier components) in those patients. Additionally, for rTLR patients, the recognition bias for fearful relative to neutral faces was attenuated with larger LPP emotion differentiation in favor of fear. This might reflect a stronger reliance of rTLR patients on cognitive processes to support fear

discrimination.^{33,47} Future work should test such effects in more detail. Indeed, there is growing evidence emphasizing an alternative to the amygdala-centric view of emotion processing. This mechanism might be, in part, realized within the visual system itself.¹⁰

And line 278 onward: “Furthermore, adequate memory representations in ITLR patients seemed dissociated from visual salience^{33,48} as shown by a missing association of EPN emotion differentiation with the memory bias.”

Discussion:

Important limitations of this study are missing: Even though 18 subjects per group are significantly more than were examined in related previous studies and recruitment of such specific patients is difficult, it should still be noted that these are still relatively small groups for EEG examination and the results should therefore be interpreted with caution.

Thank you, for pointing out this unfortunate reality of many clinical studies which we acknowledge in the manuscript. Still, our sample sizes should be, at least theoretically (Clayson et al., 2019, Psychophysiology; Jensen & MacDonald, 2022, Psychophysiology), sufficient to investigate the present measures, which we added to the methods section (l. 346):

“Nevertheless, data simulations indicate that the present participant and trial numbers should be sufficient to investigate the measures of interest.⁷⁷”

Furthermore, we now acknowledge the limitations of a clinical sample more clearly in a dedicated limitations section (l. 307 onward):

“Although these data advance our understanding of the neural mechanisms underlying emotional vision, their interpretation has some limits that need to be acknowledged. First, generalizability is limited given that our sample consisted of a clinical population. While we tried to control for between-sample confounds using a case-control study, it remains difficult to deduce the function of healthy brain tissue based on the comparison of a clinical sample with a healthy control group.”

Second, the a-priori defined channel clusters for analyzing the ERP components were based on studies of healthy control subjects. The resection-induced changes in volume conductor properties within the resection area might have led to shifts of ERP topographies. The contribution of these volume conductor related changes on ERP topographies is rather unclear but could be estimated on the basis of highly realistic head models and inverse source modelling.

Thank you for this remark. We are aware of a recent discussion on this issue about the impact of CSF cavities on the topographic distribution of EEG signals (Piai et al., 2024; Psychophysiology). Since we very much share this interest, we aim to reconstruct the EEG sources with realistic head-models in the future. However, these analyses are quite extensive and even with highly realistic head models inverse source modelling has no 100% valid unique solution and requires a considerable number of analysis choices that affect the outcome. Therefore, we currently refrain from including this approach in this manuscript. Nevertheless, since it is a very important constraint of the present study, we made sure to explicitly state this in the discussion (l. 311 onward):

“Second, scalp EEG cannot reliably uncover activity within deeper brain regions.¹⁵ Hence, to directly investigate amygdalar activity, intracranial recordings might be best suited for such analyses.³⁸ Similarly, the neural sources of the surface effects are still unknown, which future studies could approximate with inverse source modelling. Such analyses could also estimate the role of resection-

induced changes in volume conductor properties. Relatedly, large cerebrospinal fluid cavities might have led to distortions of scalp topographies.^{71,72}

Even if the restrictions due to the maximum length of the manuscript in this journal are quite strict, it would be extremely helpful if the main results of this study were placed in the context of the corresponding studies on emotional word processing (76) and the processing of emotional scenes (33), as these studies are based on predominantly overlapping subject groups. Perhaps the most important cross-stimulus results and the most important stimulus-specific differences between the studies could be reproduced here in the discussion. A more detailed discussion of this could be provided in the supplementary material.

Since we agree that this is a very interesting and above all important contextualization of our results, we added this to several paragraphs in the discussion. First, we discussed stimulus-specificity of the rTLR-P1 effect (l. 240 onward):

“Interestingly, absent emotion differentiation in the P1 in rTLR patients is seemingly not limited to facial stimuli, as it was also found for more complex emotional scenes³⁴ and more abstract written words.⁴⁴ Thus, early emotion discrimination might rely on the right amygdala across stimulus modalities. This might specifically underlie rapid, low resolution thalamo-amygdalar fear cues.^{32,45}”

Then, we also discuss the face-specificity of the changes that we found in ITLR patients (l. 280 onward):

“Of note, these mechanisms seemingly do not generalize to word processing.⁴⁴ Here, ITLR patients showed a rather consistent decrease in left-hemispheric emotion differentiation in the N1 and LPP components beyond the P1. Since word processing is thought to depend on lexical categorization in left fronto-temporal areas,⁵⁷ cerebral networks for emotion differentiation in abstract and learned stimuli such as written words could differ from those recruited during the processing of biologically salient emotion cues like faces.⁵⁸”

Methods:

"Finally, epochs containing residual artefacts (e.g., muscular activity, technical noise) that were not previously detected in the time domain were excluded from analysis after careful trial-by-trial visual inspection".

Was this artefact rejection conducted without knowledge of group membership (blinded)? If not, this should be discussed here as a limitation.

Thank you for pointing this out. It was unknown to the researcher which condition a trial belonged to, but the group was known. We tested for systematic differences in trial counts between groups and report these results a little earlier in the methods section (l. 431 onward):

“On average, 16.11 % of trials per condition were rejected. The rejection rate did not differ between conditions ($t(53) = 0.084$, $p = .933$), but marginally fewer trials were rejected in the rTLR data compared to HC ($t(35) = 2.300$, $p = .083$) and ITLR ($t(35) = 2.260$, $p = .083$). On average, 1.92 % of electrodes were interpolated per participant. The channel interpolation did not differ between groups (all $ps \geq .222$).”

We want to note here that these tests were not corrected for multiple comparisons, i.e. they are quite sensitive to any potential differences.

For transparency, we now explicitly state what was blinded and what was not in the methods section (l. 464 onward):

“This rejection was conducted without knowledge of the condition that a trial belonged to. However, it was not blind regarding the group allocation. Nonetheless, this did not lead to systematic differences in trial counts between groups (see paragraph “EEG processing”).”

Reviewer #2 (Remarks to the Author):

The authors investigated hemispheric differences in the contributions of medial temporal lobe (mTL) to fearful face processing. They compared patients with left mTL resection and right mTL resection with controls (18 participants per group). Behaviorally, no differences among groups were seen for emotion effects on face memory recognition (but generally reduced memory for both neutral and fearful faces in right TLR patients). A P1 amplitude increase for fearful versus neutral faces that was normally observed in controls (90-130 ms post-stimulus) was attenuated in rightTLR patients. Later ERP increases for fearful vs. neutral faces (N1, EPN, LPP) were abnormally amplified in lTLR patients, relative to controls, interpreted as left mTL failing to regulate fear information transmission. Finally, gamma response to neutral faces was also abnormally heightened at 95-300 ms in rTLR patients vs controls.

The study presents highly relevant data, and the results may provide important insights to human affective neuroscience. I have previously reviewed this manuscript for another journal and the authors have addressed most of my previous concerns to full satisfaction. The methodology used seems appropriate and is described in sufficient detail to allow for reproducibility. However, I still have the following comments.

We thank you for taking the time to assess the quality of our manuscript (and doing it again!) and we are very grateful for your constructive feedback. We feel that your comments greatly improved the quality of the study report. Below, you will find a point-by-point response to the concerns raised. Responses are highlighted in blue text. Major insertions to the manuscript are highlighted in yellow. We have also tried to make the writing more accessible in places and corrected some typographical errors without highlighting each minor change in wording. The analysis scripts on GitLab are updated according to the latest modifications in response to your requests.

-«Resection extent was smaller in lTLR patients compared to rTLR patients”. Although this did not have an overall impact on the results, it seems like patients’ lesions could extend into ventral temporal cortex (Fig. 1), affecting fusiform or inferior temporal gyrus. Was temporal visual cortex differentially affected for rTLR vs lTLR patients, potentially explaining that rTLR patients showed particularly weak P1 effects for fearful faces? This is important, as the main claim is that the right amygdala is not sending inputs to the visual cortex in rTLR patients.

Thank you for pointing this out. The lesions did indeed affect higher visual areas in the temporal lobes, e.g., the fusiform gyrus (FG), although only to a small extent (in some patients less than 5% of the fusiform gyrus was affected). We did check for systematic differences in fusiform volume between groups. These analyses are now reported in the control analyses in the supplement. They did indeed reveal that the fusiform resection was marginally larger in rTLR patients ($p = .054$, see paragraph S4 in the Supplement), following the pattern of the overall resection extent.

Then, to examine whether fusiform resections might have affected emotion differentiation in the patient groups, we adapted the control models and included additional resection-specific factors as control variables. To do this accurately, we finetuned those models by using subject-specific difference values (fearful-neutral) of the actual data as dependent variables. This avoided near-zero variance estimations within groups. We also omitted the general resection volume as a control factor to avoid multicollinearity between factors. The methodological approach is described in detail in the methods

section. Significant effects were reported in the main text, while full model coefficients can be found in the Supplement:

Line 643 onward: *“To explore whether any effect of emotion might be moderated by clinical factors such as the age at resection, age at epilepsy onset, time since resection, and BDI scores, we calculated the fear-neutral difference and entered them as dependent variables into our control model. Beforementioned clinical variables served as predictors. Furthermore, to test whether the extent of hippocampal¹⁰¹ and fusiform gyrus⁶¹ (FG) resections might have contributed to altered electrophysiological emotion differentiation in patients, we investigated whether the magnitude of respective resections associated with changes in differential emotion processing in the EEG measures. Resection volume was quantified by assessing the overlap between individual lesion masks and a standardized anatomical atlas. Manually constructed lesions masks (see paragraph “resection mapping”) were first resampled to match the dimensions of the Automated Anatomical Labeling¹⁰² (AAL) atlas using nearest-neighbor interpolation. A binary lesion mask was then generated to include all nonzero voxels, representing the extent of resection. To determine the resection volume of specific brain regions, the resampled lesion mask was overlaid onto the atlas, and the number of voxels corresponding to the respective region was extracted. The proportion of region-specific voxels affected by resection was then calculated relative to the total number of region-specific voxels in the atlas. All image processing and quantifications were performed using nibabel and Nilearn in Python 3.*

Predictors were centered to avoid multicollinearity. Models were individually calculated for ITLR and rTLR patients to avoid overparameterization and to ensure a more detailed description of each patient sample. This approach was identical for the ERP components and GBA. For the GBA data, we modelled data within the Group x Emotion interaction clusters identified through the cluster-based LMM approach. Full model coefficients of ERP data are documented in Supplementary Table 10. Full model coefficients of GBA data are documented in Supplementary Table 11.”

These results did not reveal an impact on the P1 effect in any group, but it did indeed reveal an association of FG resection volume and LPP emotion differentiation in rTLR patients (l. 139 onward):

“Control models revealed that larger LPP amplitudes in response to fearful than neutral faces in rTLR patients was associated with smaller fusiform resection volume ($\beta_{\text{Group (rTLR)} \times \text{ERP}} = -0.032, p = .026$). This was not observed for the other components. The ITLR group showed no effects of resection volume on electrophysiological emotion differentiation (Supplementary Table 10).

We discuss this as a potential indicator of compensatory mechanisms in rTLR patients (l. 251 onward):

“Generally, the largely preserved post-P1 emotion differentiation in the present rTLR patients is also consistent with the idea of an increasingly distributed network for threat processing over time.^{5,11} Accordingly, we found an association of fusiform resection extent and emotion differentiation in the LPP (but not earlier components) in those patients.”

-Additionally, resection encompassed complete removal of amygdala in all patients, but 5 left TLR patients had spared hippocampus. Could this also have any impact on the EEG results across groups? Did patients with spared hippocampus show a different pattern relative to the rest of their group? Emotionally salient stimuli engage amygdala-hippocampus interactions (e.g. Zheng et al., NatCommunications; 2017). This could also have had a differential impact on habituation to faces, and thus on the resulting EEG results. Again, there is no overall effect of “resection extent” on the results, but a more specific reference to the hippocampal resection, or lack thereof, could be provided.

This is a very important remark and we thank you for the literature suggestion that we incorporated now. To account for potential differences in hippocampal resection extent and to clearly acknowledge

this in the main manuscript, we added hippocampal resection volume as a control factor into our control models, analogous to the fusiform resection.

No effects of hippocampal resection volume were found on emotion differentiation in all groups and measures (see Supplementary Tables 10 and 11).

-P1: one of the main claims is that the earlier (P1) emotion response is attenuated in rTLR but unaffected for ITLR patients. However, tables S5 and S6 (and Fig. 3) are confusing in this regard, as they show no fear effect in ITLR patients either. Instead, the P1 emotion effect seen in ITLR patients is reportedly in a slightly different location: «For ITLR patients, P1 emotion differentiation was present in a forward-shifted electrode cluster (ITLR: $\beta_{\text{fearful} - \text{neutral}} = 0.016$, $p = .050$, $d = 0.410$; BioSemi channel labels: D24, D25, D31, D32, B10, B11, B14, B15). This should be better specified in Fig. 3 (left panel). It should also be specified (if that is the case) that no such effects were observed in this cluster for rTLR patients. Also, why was it shifted in ITLR patients?

Thank you for pointing this out. Reviewer 1 made a similar remark. This was indeed reported in a confusing manner, and the report of these effects was not very elegant or transparent. We now include the coefficients for each group in Supplementary Table 6. Furthermore, we clarified the rationale for this analysis in the methods section (l. 451 onward):

“Additionally, post-hoc assessment of the differential P1 topography (fearful-neutral) showed a topographic shift of the emotion differentiation in ITLR patients. Therefore, we conducted a separate analysis of P1 amplitudes in a shifted electrode cluster (BioSemi channel labels: D24, D25, D31, D32, B10, B11, B14, B15, see Supplementary Fig. 2).”

Figure 2 was updated so that the “conditionality” of the significance of this effect is now clearer, as we put parentheses around the significance mark and elaborated this in the figure caption (l. 169):

“Since significant emotion differentiation in ITLR patients was shifted forwards ($\beta_{\text{fearful} - \text{neutral}} = 0.106$, $p = .050$, $d = 0.410$; Supplementary Fig. 2) and not found in the a priori defined cluster ($\beta_{\text{fearful} - \text{neutral}} = 0.119$, $p = .277$, $d = 0.185$), the significance marking is depicted in brackets.”

We included a boxplot of the shifted electrode cluster in Supplementary Fig. 2.

Regarding the “why” – this is still speculative, but we included a tentative hypothesis regarding the underlying mechanism in the discussion. This is based on a previously published report (Kissler et al., 2023; LCN) (l. 275 onward):

“ITLR might also lead to re-organizational processes that affect early visual perception as reflected in the topographically shifted P1 emotion differentiation. A similar topographic shift in the P1 was found for those patients during word processing.⁴⁴”

Minor:

-In Table 2, it is said: «In the STAI, both patient groups had marginally lower trait anxiety scores compared to the HC group» but the table shows the opposite pattern (higher STAI trait scores for patients vs. healthy controls).

We apologize for this oversight and changed the statement accordingly.

-Please revise p-values, as one p-value in Table S6 is negative, and some p-values reported in the main text (e.g. page 6) do not match those reported in Table S5.

Thank you for catching this error. We now aligned the p-values from the text with those depicted in the tables, and we double-checked all analyses to make sure the values in the tables are correct. Just to

clarify: After changing to non-parametric tests to ensure comparability between ERP and GBA analyses, we apparently forgot to change the model p-values from parametric to non-parametric in the main text, sorry!

ACCEPTANCE OF MANUSCRIPT **COMMSBIO-25-0793**

Reviewer #1 (Remarks to the Author):

The authors have discussed and dealt with all comments and questions to my complete satisfaction. I would like to congratulate you on this excellent manuscript. I would recommend that this paper be accepted in its present form.

Reviewer #2 (Remarks to the Author):

I thank the authors for thoroughly addressing all my remaining concerns. The manuscript has improved considerably and I have no further comments.

Response to both reviewers: Again, we would like to emphasize that we are very grateful for your helpful and constructive feedback. Your expertise has much improved the quality of the manuscript. Thank you for taking the time to read and review our manuscript.